# Computational analysis of missense filamin-A variants, including the novel p.Arg484Gln variant of two brothers with periventricular nodular heterotopia

**Umut Gerlevik**[1☯¤], **Ceren Saygı**[2☯], **Hakan Cangül**[3], **Aslı Kutlu**[1,4], **Erdal Fırat Çaralan**[3], **Yasemin Topçu**[5], **Nesrin Özören**[2], **Osman Uğur Sezerman**[1,6]*

1 Department of Biostatistics and Bioinformatics, Institute of Health Sciences, Acıbadem Mehmet Ali Aydınlar University, Istanbul, Turkey, 2 Department of Molecular Biology and Genetics, Boğaziçi University, Istanbul, Turkey, 3 Center for Genetic Diagnosis, Istanbul Medipol University, Istanbul, Turkey, 4 Bioinformatics & Genetics, Faculty of Engineering and Natural Science, İstinye University, İstanbul, Turkey, 5 Department of Pediatric Neurology, Faculty of Medicine, Istanbul Medipol University, Istanbul, Turkey, 6 Department of Biostatistics and Medical Informatics, School of Medicine, Acıbadem Mehmet Ali Aydınlar University, Istanbul, Turkey

☯ These authors contributed equally to this work.
¤ Current address: Department of Biochemistry, University of Oxford, Oxford, United Kingdom
* ugur.sezerman@acibadem.edu.tr

**Data Availability Statement:** Data generated by molecular dynamics simulations and the related configuration files are available at Zenodo (doi: 10.

## Abstract

### Background

Periventricular nodular heterotopia (PNH) is a cell migration disorder associated with mutations in Filamin-A (*FLNA*) gene on chromosome X. Majority of the individuals with PNH-associated *FLNA* mutations are female whereas liveborn males with *FLNA* mutations are very rare. Fetal viability of the males seems to depend on the severity of the variant. Splicing or severe truncations presumed loss of function of the protein product, lead to male lethality and only partial-loss-of-function variants are reported in surviving males. Those variants mostly manifest milder clinical phenotypes in females and thus avoid detection of the disease in females.

### Methods

We describe a novel p.Arg484Gln variant in the *FLNA* gene by performing whole exome analysis on the index case, his one affected brother and his healthy non-consanguineous parents. The transmission of PNH from a clinically asymptomatic mother to two sons is reported in a fully penetrant classical X-linked dominant mode. The variant was verified via Sanger sequencing. Additionally, we investigated the impact of missense mutations reported in affected males on the FLNa protein structure, dynamics and interactions by performing molecular dynamics (MD) simulations to examine the disease etiology and possible compensatory mechanisms allowing survival of the males.

5281/zenodo.4483108). All codes used in the analysis are available in the GitHub repository (https://github.com/ugerlevik/FLNA_analysis). dbSNP138. https://www.ncbi.nlm.nih.gov/snp/. Accessed 1 May 2020. ExAC Browser. http://exac. broadinstitute.org/. Accessed 1 May 2020. PubMed. https://pubmed.ncbi.nlm.nih.gov/. Accessed 1 May 2020. OMIM. https://www.omim. org/. Accessed 1 May 2020. HGVS. http:// varnomen.hgvs.org/. Accessed 7 Dec 2020. GenBank. https://www.ncbi.nlm.nih.gov/genbank/. Accessed 7 Dec 2020. RCSB PDB. http://www. rcsb.org/. Accessed 7 Dec 2020. ClinVar. https:// www.ncbi.nlm.nih.gov/clinvar/. Accessed 7 Dec 2020.

**Funding:** The author(s) received no specific funding for this work.

**Competing interests:** The authors have declared that no competing interests exist.

**Abbreviations: ABD**, Actin-binding domain; **ABS**, Actin-binding sequence; ***ARFGEF2***, ADP ribosylation factor guanine nucleotide exchange factor 2 gene; **CEACAM1**, Carcinoembryonic antigen-related cell adhesion molecule 1 protein; **CH**, Calponin-homology; **DNA**, Deoxyribonucleic acid; ***FLNA***, Filamin-A gene; **FLNa**, Filamin-A protein; **FLNb**, Filamin-B protein; **FLNc**, Filamin-C protein; **H-bond**, Hydrogen bond; **Ig**, Immunoglobulin-like; **MD**, Molecular dynamics; **mRNA**, Messenger ribonucleic acid; **PC**, Principal component; **PCA**, Principal component analysis; **PDB**, Protein data bank; **PNH**, Periventricular nodular heterotopia; **PPI**, Protein-protein interaction; **RalA**, Ras-related protein Ral-A; **Rg**, Radius of gyration; **RMSD**, Root-mean-square deviation; **RMSF**, Root-mean-square fluctuations; **SASA**, Solvent accessible surface area; **WES**, Whole exome sequencing.

## Results

We observed that p.Arg484Gln disrupts the FLNa by altering its structural and dynamical properties including the flexibility of certain regions, interactions within the protein, and conformational landscape of FLNa. However, these impacts existed for only a part the MD trajectories and highly similar patterns observed in the other 12 mutations reported in the liveborn males validated this mechanism.

## Conclusion

It is concluded that the variants seen in the liveborn males result in transient pathogenic effects, rather than persistent impairments. By this way, the protein could retain its function occasionally and results in the survival of the males besides causing the disease.

## Introduction

Periventricular nodular heterotopia (PNH, OMIM [1]: 300049) is a disorder of neuronal migration which occurs during fetal development [2]. PNH is defined by lining immigrant neuron nodules in the periventricular surface of the brain [3]. It is the primary reason of drug-resistant epilepsy, and its main feature is seizures. PNH is associated with Filamin-A (*FLNA*) and ADP ribosylation factor guanine nucleotide exchange factor 2 (*ARFGEF2*, OMIM 608097) mutations, and chromosome 5p duplications (OMIM 608098) [4]. When the others are autosomal recessive, genetic basis of the *FLNA*-related PNH is X-linked dominant, and it is firstly reported in 1998 [5].

*FLNA* gene has 48 exons and encodes ~280.7 kDa Filamin-A (FLNa) protein. FLNa is a member of a three-protein family with similar structural features (i.e., types of the domains and number of the domains): FLNa (ubiquitously expressed, but it is the main type in brain and blood vessels), FLNb (ubiquitously expressed, but it is the dominant one in bones) and FLNc (expressed in skeletal and cardiac muscles) [6–8]. FLNa functions in a V-shape dimer form and contains an actin-binding domain (FLNaABD), rod 1 domain (that contains immunoglobulin-like (IgFLNa) domain repeats 1–15), two hinge domains, and rod 2 domain (that includes IgFLNa repeats 16–24) as shown in Fig 1. Most of these repeats are known to be participating in protein-protein interactions (PPIs), and there are more than 90 known interaction partners of FLNa [9]. Some of these PPIs play roles in different pathways related to actin cytoskeleton-associated functions such as cell shape, adhesion and migration [9]. FLNaABD, IgFLNa4, IgFLNa21 and IgFLNa24 are four of the domains related to the action cytoskeleton-associated functions, and variants on these domains are previously found in PNH. Since the variants on these domains were previously studied in the literature, the structural and functional details about them are shared in the following paragraphs.

FLNaABD consists of two calponin-homology domains (i.e., CH1 and CH2) with a linker (Ser151-Glu162) between them, and FLNaABD crosslinks actin filaments functioning in cell shape and migration [10, 11]. FLNaABD exists in an open and a closed state which are characterized by the accessibility of its actin-binding sequences (ABSs). Studies showed that there are three ABSs, but there is no consensus regarding the exact boundaries: ABS1 (Asn47-His56), ABS2 (Leu121-His147) and ABS3 (Ala164-Leu177) [10]; and ABS1 (Gln46-Glu55), ABS2 (Val130-Ser149) and ABS3 (Leu173-Trp192) [12]. A study involving a structure of FLNaABD-actin complex and functional studies [11], demonstrated that important regions for actin

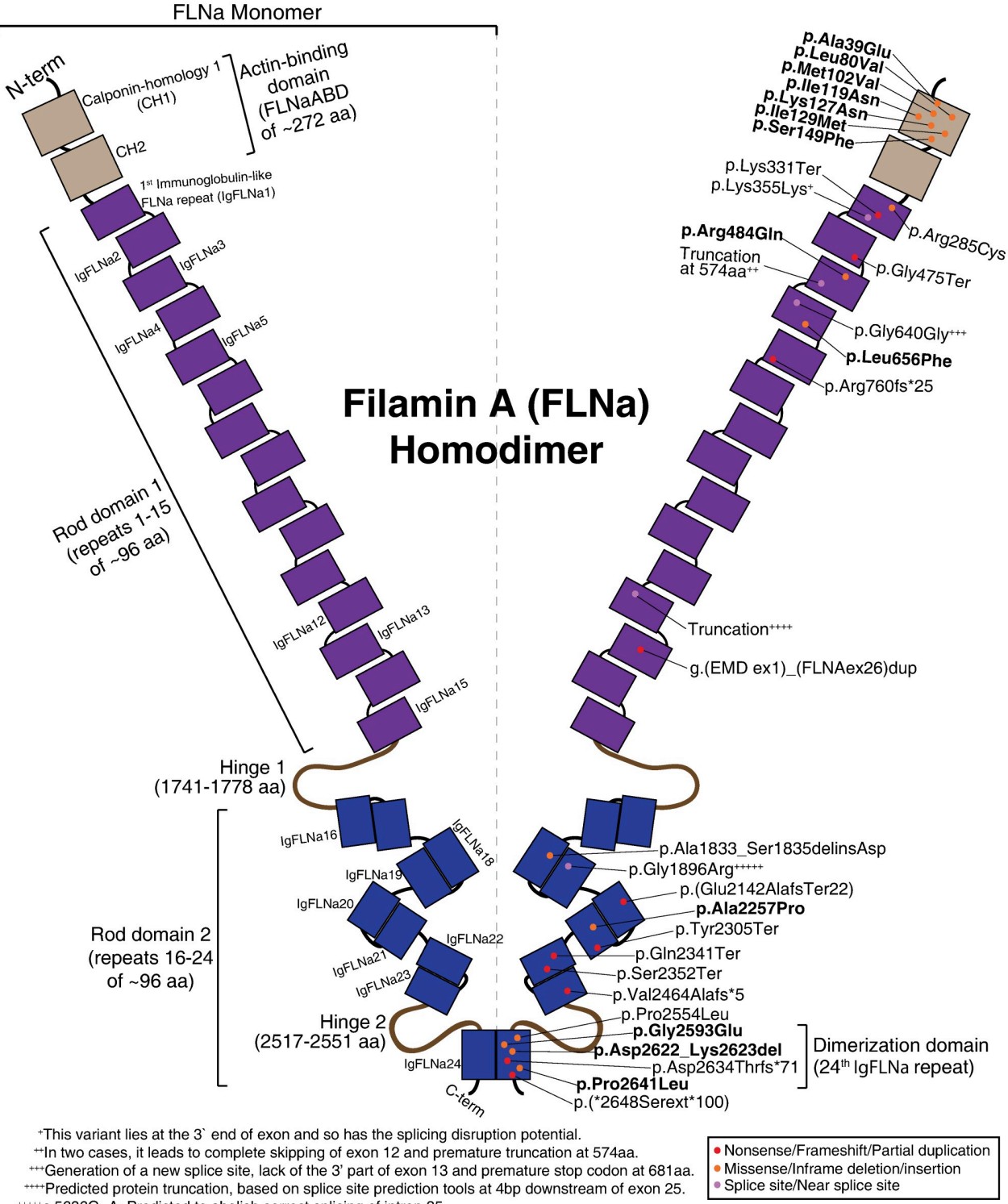

**Fig 1. Schematic representation of the Filamin-A (FLNa) homodimer.** The structural and functional domains, and the reported liveborn PNH male variants in the literature were represented on associated region of the structure. The variants subjected to molecular dynamics simulations are shown in bold. †aa: amino acid.

targeting and binding are ABS-N (Leu35-Lys42), ABS2 (V122-W142) and ABS2' (Arg91-Leu104). Movement of helix-A (Pro40-Val60) could be significant for actin-binding [11]. Also, the linker might be important for actin-binding by influencing the accessibility of ABSs [12].

IgFLNa21 has numerous known interaction partners. Integrin is one with crucial roles in cell migration by interacting with talin [13]. IgFLNa21-integrin interaction plays a negative regulatory role by keeping the integrin away from talin [14]. Migfilin is another partner that uses the same binding site (Gly2269-Phe2285) as integrin. Migfilin regulates the integrin-IgFLNa21 interaction by competing with integrin to bind IgFLNa21 [6, 9, 15]. Another regulatory mechanism of IgFLNa21 is auto-inhibition by adjacent domains. Specifically, the binding site in IgFLNa21 could be inhibited by IgFLNa20 upon domain rearrangements [6, 16]. Since IgFLNa4 is highly similar to IgFLNa21 according to Ithychanda *et al.* (2009), it can replace IgFLNa21 or simultaneously bind the ligands [6]. According to their model [6], the corresponding ligand-binding site in IgFLNa4 is Gly607-Cys623.

IgFLNa24 is responsible for the dimerization of FLNa, which results in its functional form. The dimerization interface is Asn2587-Gly2610 [17]. Also, Arg2612 and Tyr2614 are in the interface if the structure is carefully examined (S1A Fig). Also, three residues (i.e., Leu2591, Val2592 and Gly2593) significantly contribute to the dimerization by hydrophobic stacking [17]. IgFLNa24 has many known interaction partners. One of them is Ras-related protein Ral-A (RalA). Their interaction facilitates connection of RalA to actin cytoskeleton to form filopodia for cell motility [18, 19]. Carcinoembryonic antigen-related cell adhesion molecule 1 (CEACAM1) interaction with IgFLNa24 negatively regulates the complex with RalA, and so its binding to the IgFLNa24 reduces the cell migration [20]. For both partners, the binding sites are unknown. However, the putative phosphorylation site, Ser2640-Pro2641-Tyr2642, might be a regulatory switch for these interactions [17].

Disease mechanism of *FLNA*-associated PNH is known as loss-of-function [3, 7, 9, 21]. FLNa mutations presumably disrupt the FLNa-partner interactions related to cell migration or the linked functions (e.g., filopodia formation) through directly from the binding sites or indirectly from a regulatory site. Up to now, more than 130 *FLNA* mutations have been reported in the patients with PNH. They show different clinical outcomes in male and female patients. Many female patients are asymptomatic or have only mild symptoms, and they do not show partial epilepsy and mental retardation as observed in the males [22–24]. However, liveborn male reports are rare in the literature because the X-linked *FLNA* mutations are mostly lethal for males as suggested upon the common miscarriages, premature male deaths of affected mothers and skewed sex ratio in the families [22, 25–28].

The first living male with a *FLNA* mutation was reported in 2001 [29]. Since that day, only 41 affected males with 32 different *FLNA* mutations have been published, including the two males reported in this study [29–47]. All these mutations are summarized in S1 Table and shown on a schematic FLNa structure in Fig 1. *FLNA*-associated PNH is predominantly seen in women with difficult to treat seizures. In contrast, hemizygous *FLNA* mutations in males are mostly lethal, and it is assumed that the loss of function mutations in males results in a more severe phenotype [4, 5, 48]. There are two different scenarios to explain the liveborn males with *FLNA* mutation: (1) mosaicism and (2) partial loss of function. There are several reports that revealed the mosaicism of *FLNA* in males with a classical PNH phenotype [30, 40]. Hehr *et al.* (2006) reported a male with a splice site mutation in *FLNA*. They showed that the splice site resulted in both normal and aberrant mRNA transcripts and this can result by retaining normal *FLNA* function partially [42]. There are other studies which points out the same mechanism, 7 of 41 known affected males were also reported with the expression of both normal and aberrant mRNA transcripts [30, 34, 39, 40, 44, 46, 47]. In addition, mutations in

*FLNA* consistent with residual function were reported to cause PNH in males, with a less severe outcome; often missense changes, or alleles that only truncate the extreme C-terminus [29, 49, 50].

Heterodimerization of FLNa with FLNb could be another explanation to liveborn males [51] despite the study by Himmel *et al*. [52] which is opposed to the heterodimerization. FLNa probably functions in the initiation of cell migration instead of the maintenance of its rate [3, 21, 51, 53]. Moreover, Sheen *et al*. (2002) proposed that heterodimerization might sufficiently save the initiation. Even though these mechanisms might explain the functional recovery, there could be other unknown mechanisms.

Herein, we report on the transmission of PNH from a clinically asymptomatic mother to two sons, in a fully penetrant classical X-linked dominant mode. We identified a novel c.1451G>A change in FLNA exon 10, leading to the substitution of a highly conserved amino acid (p.Arg484Gln). Moreover, we performed molecular dynamics (MD) simulations to examine the impact of the novel variant and most of the missense FLNa variants reported in the liveborn males with PNH. Our aim is to elucidate the PNH etiology in males, and to understand how the function might be compensated.

## Methods

### Identification of the novel mutation in *FLNA*

The index patient is a 16-year-old male with drug-resistant occipital lobe epilepsy and epileptic status in sleep. Pedigree demonstrated that there is one additional 11-year-old affected brother and healthy non-consanguineous Turkish parents (Fig 2). Furthermore, this was an unresolved rare disease case for years since the routine whole exome sequencing (WES) analysis did not provide a diagnosis. Thus, we considered to reanalyze the data. While collecting the sample, since this was a long-time undiagnosed case, the physician informed the patient's family and obtained the written consent form allowing us to use the data for further research and publication. They volunteered the with the hope of getting diagnosed and the data can be used for diagnosis of other children as well. Therefore, we did not obtain the data as part of a research study which would require ethical board approval.

Genomic DNA was isolated from leukocytes in peripheral blood samples of the family members. Captured libraries were loaded onto the NextSeq 500 platform (Illumina, San Diego, California, USA). Trimmomatic [54] was used to remove adapter sequences and sequences of low quality bases. Further processing was performed with the Genome Analysis Toolkit's (GATK v3.4) [55], following the best practice recommendations of the developers. Briefly, trimmed reads were aligned to the human reference assembly (UCSC GRCh37/hg19) using the Burrows-Wheeler Aligner (BWA mem v0.7.12) [56]. Picard (v1.141, http://broadinstitute.github.io/picard, Cambridge, Massachusetts, USA) was used to remove the duplicate reads. GATK was used for indel realignment, base quality score recalibration, calling variants using the HaplotypeCaller, joint genotyping, and variant quality score recalibration. Afterward, variants were annotated and compared to those documented in publicly accessible genetic variant databases (e.g., dbSNP138 [57], ClinVar [58], ExAC [59], and gnomAD [60]) using AnnoVar (v2015-03-22) [61]. The variant call format (vcf) files were annotated using AnnoVar. and then integrated with OMIM phenotypes, HPO terms.

Only exonic and splicing variants that were absent in public databases, rare (with a minor allele frequency of ≤ 0.1% and no homozygotes in public databases), or located at exon-intron boundaries ranging from −40 to +40 were retained. The variants were then prioritized by pathogenicity predictions using multiple *in silico* tools (CADD [62], REVEL [63], M-CAP [64], PrimateAI [65], SIFT [66], PolyPhen2 [67], and MutationTaster [68]). Effect of the splicing

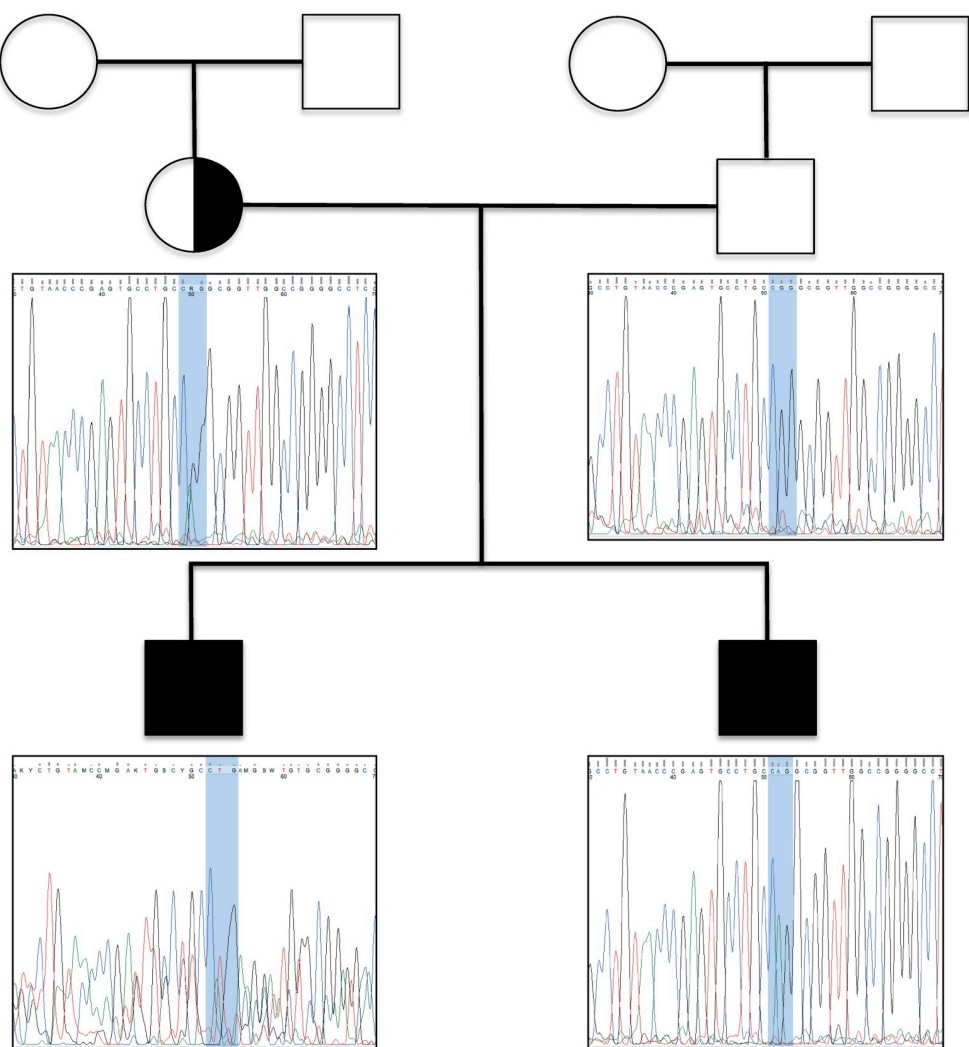

**Fig 2. Pedigree and electropherograms of a non-consanguineous Turkish family segregating X-linked dominant periventricular nodular heterotopia.** Two hemizygous brothers carry the novel c.1451G>A (p.Arg484Gln) variant. Asymptomatic mother is a heterozygous carrier of the variant. Father carries the reference allele.

variants were also investigated using four different splice site prediction programs (Human Splicing Finder [69], NetGene2 Server [70], Berkeley Drosophila Genome Project-Splice Site Prediction by Neural Network [71] and Oriel SpliceView [72, 73]). Then, the remaining variants were further prioritized based on literature and predicted effects on protein function. The detailed variant filtering strategy is outlined in S2 Table.

In general, the variants should be absent in the healthy parents in *de novo* manner or present in the heterozygous state in the healthy parents for homozygous or compound heterozygous manners. WES in two affected males and their parents identified 21 variants for *de novo* heterozygous, 19 variants for homozygous and 2 variants for compound heterozygous scenarios. Further prioritization ended up with five genes namely *SULT1C3*, *SLAIN1*, *ZFHX3*, *TKTL1* and *FLNA*. Published articles in the literature related to these five genes were compiled at this stage. Known functions, domains and motifs of each protein; distribution of mutations on each gene, associated diseases, the inheritance patterns of the diseases, exceptions regarding the inheritance patterns, overlapping and differentiating symptoms in affected individuals

were all collected to further examine possible correlations of these genetic variants with the clinical features of the two patients. Among all, *FLNA* was the gene that can be associated with the disease due to the highly-matched symptoms, X-linked dominant mode of transmission pattern, and the results of molecular dynamics (MD) simulations.

Sequence validation and segregation analysis was performed by Sanger sequencing. Detailed primer sequences and PCR conditions are available upon request. Sequence electropherograms were analyzed using the FinchTV (Geospiza, Inc.; Seattle, WA, USA; http://www.geospiza.com). Mutation nomenclature refers to GenBank [74] mRNA reference sequence NM_001110556.2. HGVS nomenclature notation [75] is used to state the variants. The variant found in this family is available under ClinVar [58] accession SCV001437633.

## Evolutionary conservation analysis

Evolutionary conservation of the variants among *FLNA* orthologues was examined by using T-Coffee [76]. Potential functional importance of the variations is supported by the fact that the region is quite conserved in several species: human (*Homo sapiens*, NP_001104026.1), rhesus macaque (*Macaca mulatta*, XP_001091073.1), dog (*Canis lupus familiaris*, XP_867483.1), bovine (*Bos taurus*, NP_001193443.1), mouse (*Mus musculus*, NP_034357.2) and brown rat (*Rattus norvegicus*, NP_001128071.1).

## Molecular dynamics studies

**Variant list.** In the literature, we could have identified 32 different *FLNA* variants reported in the males with periventricular nodular heterotopia (PNH). They are summarized in S1 Table and shown on the FLNa schematic structure in Fig 1. We created a subset by keeping only the small in-frame deletions/insertions and missense variants to make a suitable list to study their impact on the FLNa structure via MD simulations. We only selected the variants whose sites are resolved in in the RCSB database [77]. Final list of the simulated variants is given in Table 1.

**System preparation.** Four different crystal structures were used with PDB IDs: 3hop [10], 4m9p [78], 2brq [13] and 3cnk [17]. The structures from 3hop and 4m9p are monomers of

**Table 1. The filamin-A (FLNa) variants reported in the liveborn male patients with periventricular nodular heterotopia.** The full list given in S1 Table was filtered into this list to be able to study the impact of the variants via molecular dynamics simulations.

| Amino acid alterations | Domain | PDB ID | PDB reference |
|---|---|---|---|
| p.Ala39Glu | Acting-binding domain (FLNaABD) | 3hop | Clark *et al.*, 2009 |
| p.Leu80Val | | | |
| p.Met102Val | | | |
| p.Ile119Asn | | | |
| p.Lys127Asn | | | |
| p.Ile129Met | | | |
| p.Ser149Phe | | | |
| **p.Arg484Gln** | IgFLNa3 | 4m9p | Sethi *et al.*, 2014 |
| p.Leu656Phe | IgFLNa4 | | |
| p.Ala2257Pro | IgFLNa21 | 2brq | Kiema *et al.*, 2006 |
| p.Gly2593Glu | IgFLNa24 (dimerization domain) | 3cnk | Seo *et al.*, 2008 |
| p.Asp2622_Lys2623del | | | |
| p.Pro2641Leu | | | |

[†] "Ig" refers to "immunoglobin-like" (e.g., IgFLNa3 stands for immunoglobulin-like FLNa domain repeat 3.).

**Table 2. Approximate dimensions and atom count of the systems prepared for molecular dynamics simulations, and simulation time lengths.**

|  | Dimensions on xyz (Å) | Atom count | Simulations |
|---|---|---|---|
| **3hop Systems** | $94 \times 70 \times 75$ | 47,000 | 100 ns (original), 100 ns (repeat) |
| **4m9p Systems** | $75 \times 75 \times 94$ | 49,000 | |
| **2brq Systems** | $69 \times 96 \times 76$ | 47,000 | |
| **3cnk Systems** | $71 \times 74 \times 69$ | 33,000 | |

FLNaABD and IgFLNa3-5, respectively. 3cnk is a homodimer of IgFLNa24 whereas 2brq is a homodimer of IgFLNa21-integrin β-7 cytoplasmic tail complex. These states were preserved in our systems.

The p.Asp2622_Lys2623del mutant was modelled by using SWISS-MODEL [79]. Only the atoms of FLNa were kept whereas the water and smaller molecules related to the crystallization procedure were removed from the crystal structures. Each variant was modelled in a separate system. The structures were solvated in TIP3P water [80] boxes and neutralized by adding NaCl with cut-off distance of 5 Å. All procedures were performed by using Visual Molecular Dynamics 1.9.3 (VMD) [81]. The approximate system sizes and dimensions are given in Table 2.

**Simulation setup.** By using NAMD 2.13-multicore-CUDA [82], all systems were subjected to MD simulations with CHARMM36m all-atom additive force field [83]. Following a 10,000-step minimization with conjugate gradient algorithm, they were equilibrated for 1 ns at 298 K under NVT ensemble. Production simulations (original) were performed along 100 ns at 310 K and 1 atm under NPT ensemble. All production simulations were run twice along 100 ns by changing the random number seeds (repeat) and so the assigned velocities from the Boltzmann distribution to search more conformations. For pressure and temperature controls during simulations, Nosé-Hoover Langevin barostat [84, 85] and Langevin thermostat [86] were employed. ShakeH algorithm of NAMD was applied for water molecule constraints during all the simulations. 12 Å cut-off distance was used for van der Waals interactions. Switching function starts at 10 Å and reaches zero at 14 Å. Integration time-step was 2 fs. While computing the long-range Coulomb interactions, the particle-mash Ewald [87] method was used.

**Trajectory analysis.** Loop/linker flexibilities, hinges and conformational states are important for the PPIs. Overall protein folding, compactness and solvent exposure measures are also important for the proper PPIs besides the stability of the structure. In-house tcl scripts and built-in plugins of VMD were used to calculate aligned backbone root-mean-square deviation (RMSD; indicating how protein folding changes), average root-mean-square fluctuations (RMSF; referring to how flexibility changes) of aligned $C_\alpha$ atoms, radius of gyration (Rg; inversely related to how compactness of the protein alters), solvent accessible surface area (SASA; indicating how much area is exposed), distances, and number of interactions. Local impacts were searched within 10 Å of each mutation site. Hinge sites were accessed at each $10^{th}$ ns from the slowest non-trivial mode of Gaussian network models [88] with 10 Å distance cutoff and 1.0 gamma value, built via ProDy (v.1.10.11) [89]. ProDy was also used for principal component analysis (PCA) on $C_\alpha$ atoms to obtain conformational spaces with principal components 1 and 2 (PC1 and PC2) after eliminating rotational and translational movements. We plotted PCA by using matplotlib [90] package in python (v3.7) [91]. FoldX (v5.0) [92] was used to calculate binding energy ("AnalyseComplex" command) and structural stability ("Stability" command). VMD was also used for structural visualizations. To visualize the analyses, we used ggplot2 [93] package in R (v4.0.0; Vienna, Austria) [94].

## Results

### Identification of a novel variant in *FLNA*

We identified a novel c.1451G>A variant in filamin-A (*FLNA)* exon 10, leading to the substitution of a highly conserved amino acid (p.Arg484Gln, ClinVar accession: SCV001437633). We report the transmission of periventricular nodular heterotopia (PNH) from a clinically asymptomatic mother to two sons, in a fully-penetrant classical X-linked dominant manner. The healthy mother is a heterozygous carrier of the variant, and the healthy father carries the reference allele. The index case and his affected brother are hemizygous (Fig 2). In line with clinical evaluation, the individuals were diagnosed as PNH. Since p.Arg484Gln is the novel causative variant seen in the family, it was analyzed in detail to examine its impact on the FLNa protein. The variants reported in other affected males with PNH were used to validate our results.

### Impact of the novel p.Arg484Gln variant

Arg484 is an evolutionarily well-conserved (S2 Fig) amino acid found on the IgFLNa4 interface of IgFLNa3 (Figs 3B and 5B). In our affected individuals, the charged Arg484 residue is replaced with the polar and relatively small Gln. As a result of this alteration, flexibility of two loop regions (i.e., Lys508-Glu514 and Asp624-Ser630) were reduced whereas flexibility of almost entire IgFLNa3 and IgFLNa5 (Fig 3B and S3B Fig) were slightly increased. Also, there was a minor disruption in one of the four hinge regions observed within IgFLNa3-5 (S3A Table). Observed conformations of FLNa were significantly different in a part of simulations (i.e., repeat trajectories) in the variant form as shown in Fig 4B, which indicates that there might be other energetically local minima for the conformation of FLNa that could be a cause for pathogenicity.

When we focus on the overall protein structure, there were transient and modest compactness increase (Rg decrease) in line with the RMSD increase in a part of trajectories (i.e., repeat) (S4B Fig). There was no important difference in the overall structural stability of the protein (S4B Fig). Although the structure became temporarily more compact, its SASA were distributed with higher values than the wild-type's, indicating that the change was local and in the

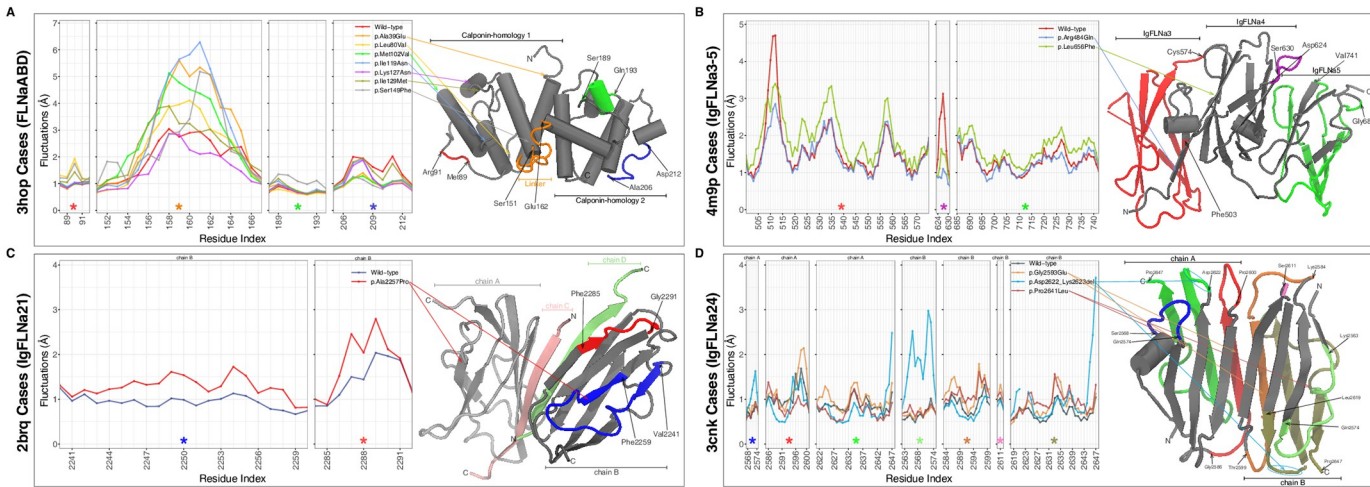

**Fig 3. Average Cα root-mean-square fluctuations (RMSF) of the amino acids during the simulations.** Only the regions whose flexibility is affected by the variants are demonstrated. They are highlighted on the cartoon representation of the structural parts of filamin-A (FLNa): (A) FLNaABD, (B) IgFLNa3-5, (C) IgFLNa21 and (D) IgFLNa24. [†]Colored stars at the bottom of the panels and the colored regions on the structure representation refer to the same region on FLNa.

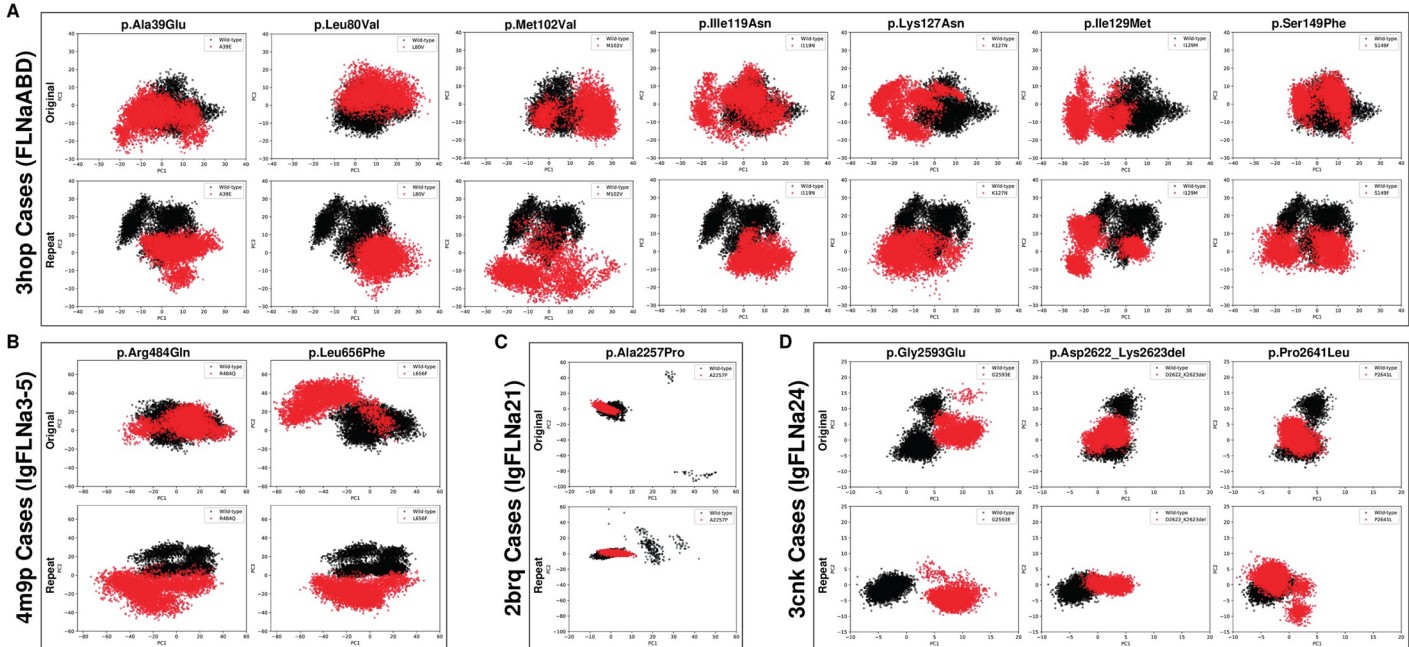

**Fig 4. Conformational spaces of the (A) FLNaABD, (B) IgFLNa3-5, (C) IgFLNa21 and (D) IgFLNa24 variants, compared to the wild-type.** The conformations collected in the molecular dynamics simulations are visualized by using the first two principal components.

buried regions (S4B Fig). In contrast to this, SASA was lower than it was in the wild-type in the other part of the trajectories (i.e., original). This was probably related to the closer ligand-binding region in the mutant which had no backbone folding change (Fig 5B).

The p.Arg484Gln variant importantly altered its local (around 10 Å) environment by decreasing the number of H-bonds (S5B Fig), and causing the ultimate disappearance of the Glu642-Arg484 interdomain salt bridge and the existence of two additional salt bridges (only in the original simulations) compared to the wild-type (S6B Fig).

### Impact of the other variants seen in males as a validation

Amino acids at the mutation sites in the liveborn males with PNH (Table 1) are well-conserved during evolution (S2 Fig). Thus, all these sites might be crucial in terms of the structure and function of FLNa. To reveal their importance, MD simulations were performed, and the structural details are explained in the followings.

All variants significantly altered the conformational spaces in a part of the original and repeat simulations (Fig 4), which shows the formation of alternative conformations of the protein structure in the variant forms, at least for a part of the simulations. Flexibility of the amino acids were affected from each mutation (Fig 3) whereas hinge points were mostly not (changes are shown in S3 Table while others are not). Number of local H-bonds were significantly altered only by the p.Leu80Val, p.Leu656Phe and p.Asp2622_Lys2623del mutations (S5 Fig). Local salt bridges were affected by most of the variations (S6 Fig). All the changes were directly or allosterically related to the protein-protein interactions (PPIs) or regulatory functions. For instance, the distance within the dimerization site was increased upon the change of side chainless Gly to a Glu, which has a large side chain and negative charge (S1 Fig). This p. Gly2593Glu variant was directly associated with PPI because the mutation site is directly on the interaction surface between FLNa monomers. Another example is the altered flexibility of

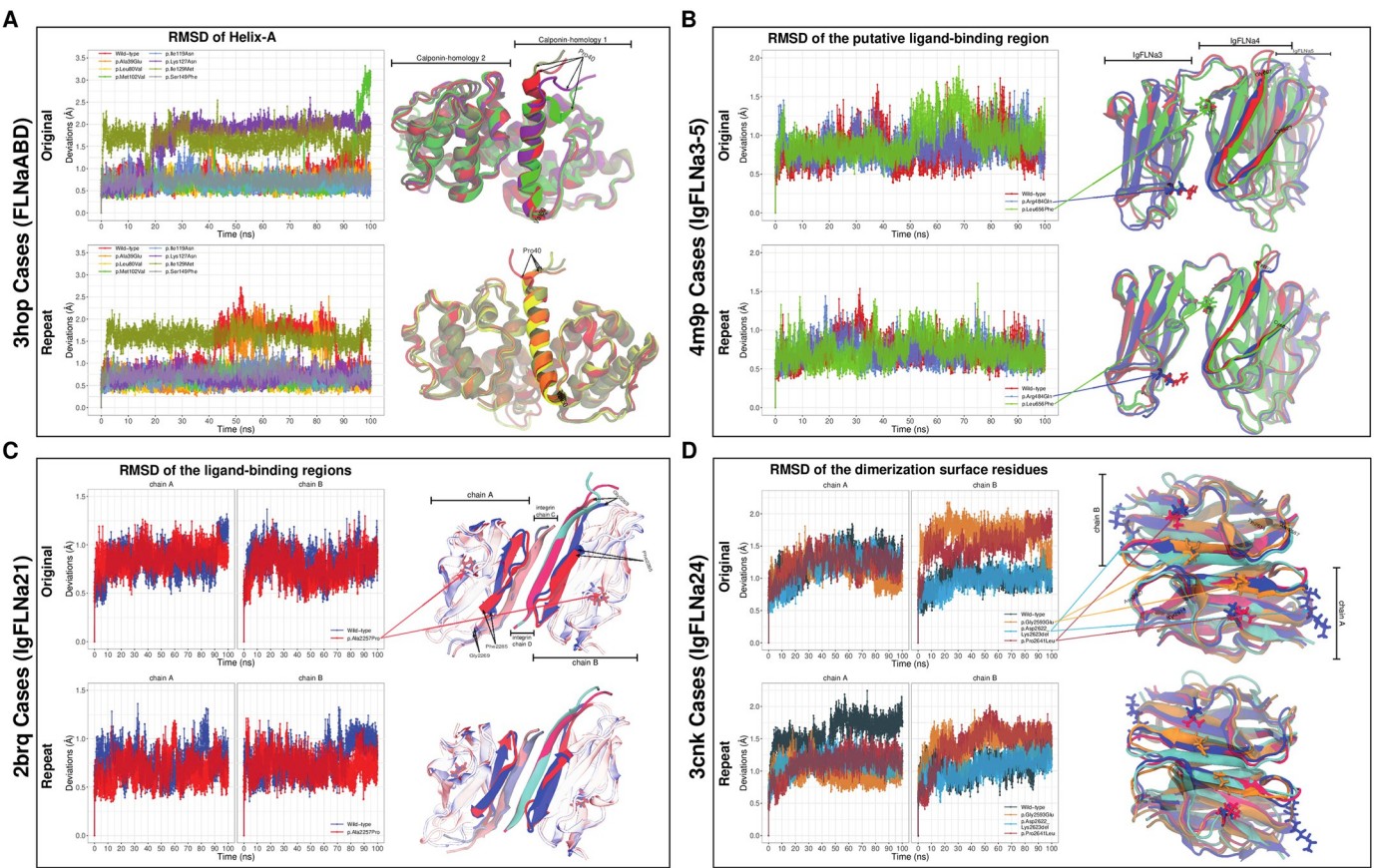

**Fig 5. Backbone root-mean-square deviations (RMSD) and visualizations of known critical regions for cell migration.** The following structural parts of filamin-A (FLNa) were represented: (A) FLNaABD, (B) IgFLNa3-5, (C) IgFLNa21 and (D) IgFLNa24. The critical regions were helix-A, a putative ligand-binding site, a ligand-binding site and dimerization interface for these parts, respectively.

a part of the FLNa-integrin interaction surface upon the p.Ala2257Pro mutation (Fig 3C), which affected the interface allosterically. However, these alternative conformations were mainly observed transiently by going back to a wild-type-like structure during the simulations.

Critical increase in flexibility of the linker region was observed for all the variants found in FLNaABD. Also, other regions, most of which contain loops, were affected by some of these mutations (Fig 3A and S3A Fig). Helix-A changes were observed in some of the mutants (Fig 5A), which might influence the actin-binding. On the other hand, there were no important changes in the overall structural stability of the protein for any of the variants found in FLNaABD (S4A Fig).

Similar to p.Arg484Gln on IgFLNa3, p.Leu656Phe on IgFLNa4 brings about a slight increase in fluctuations of entire IgFLNa3 and half of IgFLNa5 (Fig 3B and S3B Fig). As opposed to the nearing in the p.Arg484Gln, distancing of the ligand-binding region towards the IgFLNa3 might also influence the proper binding in the p.Leu656Phe mutant (Fig 5B). Additionally, flexibility of two loops, one close to the ligand-binding region, decreased as in p. Arg484Gln. Differences in hinge regions were observed in the mutant as a slight disruption in one of the hinge regions (S3A Table). There were compactness (Rg) changes in line with the RMSD and SASA for the p.Leu656Phe mutant, but there was no significant change in the over-all structural stability of the protein (S4B Fig). Both known variants (i.e., p.Arg484Gln and p. Leu656Phe) around this region were in the interface between the repeats 3 (IgFLNa3) and 4

(IgFLNa4). Therefore, they affected compactness with their altered volumes and hydrophobicity. Both variants were observed as having similar pathogenicity mechanisms.

The p.Ala2257Pro variant in IgFLNa21 caused minor flexibility increases in several loops and β-sheets including close regions to the ligand-binding site (Fig 3C and S3C Fig). However, there was no remarkable difference in the region itself (Fig 5C). There was no difference in the overall structural stability (S4C Fig).

Fluctuation levels in most of IgFLNa24, including half of the dimerization interface, were affected differently from the three variants found in there (Fig 3D and S3D Fig). For these dimerization domain variants, there were distinct patterns in other properties as well. In p. Gly2593Glu, most of the hinges were disappeared. There were also hinge region disruptions in p.Asp2622_Lys2623del whereas hinges in the p.Pro2641Leu mutant were more conserved than the wild-type (S3B Table). There were compactness changes in line with the RMSD and SASA for the p.Gly2593Glu and p.Asp2622_Lys2623del mutants (S4B and S4D Fig). There were no important differences in the overall structural stability of IgFLNa24 (S4D Fig). p. Gly2593Glu is a change from a side chainless hydrophobic residue to a charged and large amino acid in the middle of the dimerization interface. Thus, it was expected to cause a less compact dimer complex. The in-frame deletion (p.Asp2622_Lys2623del) led to a more compact complex as expected upon the shortened loop close to the interface. Changes in the number of H-bonds, distance patterns, binding energies and SASA distributions of the dimerization interface residues supported these findings and expectations (S1B Fig). The interface was also rearranged in the p.Pro2641Leu mutant although there were no alterations in interactions and binding energies (Fig 5D and S1 Fig).

## Discussion

We identified the missense variant p.Arg484Gln on filamin-A (*FLNA*) in a Turkish family, associated with periventricular nodular heterotopia (PNH). This variant was formerly submitted to ClinVar (i.e., SCV000250440.11 and SCV000639759.3) by two different submitters. However, the mutation was interpreted as variant of uncertain significance in these submissions. Also, it was not observed in an individual and not directly associated with PNH. Thus, this is the first study reporting the c.1451G>A (p.Arg484Gln) variant in *FLNA* causing PNH in the individuals.

To understand the underlying mechanism behind the pathogenicity of p.Arg484Gln, molecular dynamics (MD) simulations were performed. The simulations showed that p. Arg484Gln leads to pathogenicity by disrupting the structural and dynamic features of FLNa in several ways. Thus, these results validated the PNH diagnosis of the index patient. These impacts were observed only for a part of the simulation trajectories rather than a general and permanent disruption. This type of effect could be sufficient to disrupt the FLNa function in the initiation of cell migration whereas could it also allow to compensate of this function occasionally. By this way, the missense mutation in the single copy of *FLNA* of a male could simultaneously leads to the disease and provides functionality, and these males can survive. When we performed MD simulations on the 12 other variants of FLNa reported in males cases in the literature, we observed highly similar impact like p.Arg484Gln, which is enough for the pathogenicity but transient. Therefore, these results supported us to validate the conclusion obtained for the novel p.Arg484Gln mutation.

Some of the examined mutations in this study or other variants found at the same positions have been formerly studied in the literature. Moreover, these previous studies supported our findings. p.Ala39Gly (same position as p.Ala39Glu), p.Met102Val, p.Lys127Asn and p. Ile119Asp (same position as p.Ile119Asn) do not lead to a general misfolding, and they are also

related to strongly-impaired F-actin targeting and binding [11]. The etiology for the p. Ser149Phe mutation is unclear since it only has a little effect on F-actin targeting [11]. Ruskamo and Ylänne (2009) concluded as the p.Ser149Pro could affect the linker [12] as we observed for p.Ser149Phe in the MD simulations. We observed distancing IgFLNa24 monomers but no separation in the p.Gly2593Glu variant. Compatibly, one study showed the abolishment of homodimerization in the isolated IgFLNa24 while no disruption in the structure of IgFLNa16-24 domains [33].

Sheen *et al.* [51] proposed that heterodimerization of FLNa with FLNb might save the initiation of the cell migration process and so provide a partial function in the males with FLNa-associated PNH. Thus, the relationship between FLNa and FLNb is very important issue to be explored. Affected individuals with FLNb variants reported only with homozygosity and compound heterozygosity. The protein function is compensated by the wild-type allele (FLNb on the chromosome 3) in heterozygous carriers and they generally do not show severe symptoms [95]. In FLNa variants, this situation is valid only for females, but not for males due to hemizygosity (FLNa on the chromosome X). In several studies regarding FLNb variants [95, 96], severe pathogenic impacts on the protein's structure and stability were observed. However, in FLNa, those impacts were transient and milder than observed in FLNb, which make sense to provide an environment that is suitable for both the disease and partial function for the survival.

The limitations of this study are as follows. Firstly, the structure of FLNa has not been fully resolved by crystallography or any other structural method and building a full structural model is impossible due to the very high number of atoms. Thus, we used these four different PDB structures that cover the variants we listed. Moreover, among all known missense and small in-frame deletion mutants, MD simulations could not be performed for p. Ala1833_Ser1835delinsAsp, p.Pro2554Leu and p.Arg285Cys due to lack of the structure in the PDB database. On the other hand, FLNa structures complexed with its cell migration-associated interaction partners could be useful to further understand the impact of these variations. Such structures are mostly lack, low quality or involving mutations which retains us from performing reliable simulations for those. Despite these limitations, we think that this study provides significant insights about disease etiology in males with *FLNA*-related PNH.

## Conclusions

In this study, the novel p.Arg484Gln (c.1451G>A) variant on filamin-A (*FLNA*) was identified in a male patient with periventricular nodular heterotopia (PNH) by performing whole exome sequencing analysis. Furthermore, we aimed to elucidate the PNH etiology and the mechanisms allowing the survival of males with only one mutant copy (i.e., hemizygous males) of *FLNA* by performing molecular dynamics simulations. We concluded that the variants seen in the liveborn males result in transient pathogenic effects on the analyzed FLNa protein domains, rather than persistent impairments that might not allow a partial function, which is required for their survival with the only copy of *FLNA* gene. This situation provides an explanation for the co-existence of PNH disorder and FLNa function in those hemizygous males.

## Supporting information

**S1 Fig.** (A) Visualization of the dimerization interface in IgFLNa24 of filamin-A (FLNa), including the residues Arg2612 and Tyr2614. (B) Distance between the IgFLNa24 monomers through the time, binding energy distributions between the monomers, distributions of

number of H-bonds and solvent-accessible surface area (SASA) in the dimerization interface during the molecular dynamics simulations.
(TIF)

**S2 Fig. Evolutionary conservations of the amino acids at the mutation sites of the missense variants found in the liveborn males with periventricular nodular heterotopia.** The filamin-A orthologs from human (*Homo sapiens*, NP_001104026.1), rhesus macaque (*Macaca mulatta*, XP_001091073.1), dog (*Canis lupus familiaris*, XP_867483.1), bovine (*Bos taurus*, NP_001193443.1), mouse (*Mus musculus*, NP_034357.2) and brown rat (*Rattus norvegicus*, NP_001128071.1) were used.
(TIF)

**S3 Fig. Average Cα root-mean-square fluctuations (RMSF) of each residue at (A) FLNaABD, (B) IgFLNa3-5, (C) IgFLNa21 and (D) IgFLNa24 during the molecular dynamics simulations.**
(TIF)

**S4 Fig. Overall root-mean-square deviations (RMSD), radius of gyration (Rg; inversely-related to compactness), FoldX stability and solvent-accessible surface area (SASA) of the (A) FLNaABD, (B) IgFLNa3-5, (C) IgFLNa21 and (D) IgFLNa24 along the molecular dynamics simulations of each wild-type and mutant system.** The results are plotted as against the time (RMSD and Rg) or as the distribution of collected values during the MD simulations (stability and SASA).
(TIF)

**S5 Fig. Distribution of number of (local) H-bonds within 10 Å of the mutation sites along the molecular dynamics simulations for the variants on (A) FLNaABD, (B) IgFLNa3-5, (C) IgFLNa21 and (D) IgFLNa24, and for the corresponding wild-type systems.**
(TIF)

**S6 Fig. Distance for the affected (local) salt bridges within 10 Å of the mutation sites against the simulation time for the variants on (A) FLNaABD, (B) IgFLNa3-5, (C) IgFLNa21 and (D) IgFLNa24, and for the corresponding wild-type systems.**
(TIF)

**S1 Table. All periventricular nodular heterotopia-associated filamin-A variants reported in the liveborn males.** [†]*EMD* is the gene encoding Emerin protein.
(DOCX)

**S2 Table. Classes of evidence in our variant prioritization strategy.**
(DOCX)

**S3 Table. Predicted hinge sites in the slowest non-trivial mode of Gaussian network models along the molecular dynamics trajectories for the mutations on (A) IgFLNa3-5 (PDB ID: 4m9p) and (B) IgFLNa24 (PDB ID: 3cnk).** Hinges were evaluated based on the existence within regions defined by distance in the sequence.
(XLS)

## Acknowledgments

We are grateful to the family for participating in this study. The molecular dynamics calculations reported in this study were fully performed at TUBITAK ULAKBIM, High Performance and Grid Computing Center (TRUBA resources).

## Author Contributions

**Conceptualization:** Umut Gerlevik, Ceren Saygı, Aslı Kutlu, Osman Uğur Sezerman.

**Data curation:** Umut Gerlevik, Ceren Saygı, Aslı Kutlu.

**Formal analysis:** Umut Gerlevik, Ceren Saygı, Aslı Kutlu.

**Investigation:** Umut Gerlevik, Ceren Saygı, Aslı Kutlu, Erdal Fırat Çaralan.

**Methodology:** Umut Gerlevik, Ceren Saygı, Aslı Kutlu.

**Project administration:** Osman Uğur Sezerman.

**Resources:** Hakan Cangül, Yasemin Topçu, Osman Uğur Sezerman.

**Software:** Umut Gerlevik, Ceren Saygı, Aslı Kutlu.

**Supervision:** Hakan Cangül, Nesrin Özören, Osman Uğur Sezerman.

**Validation:** Umut Gerlevik, Ceren Saygı, Aslı Kutlu, Erdal Fırat Çaralan.

**Visualization:** Umut Gerlevik, Ceren Saygı.

**Writing – original draft:** Umut Gerlevik, Ceren Saygı, Aslı Kutlu.

**Writing – review & editing:** Umut Gerlevik, Ceren Saygı, Hakan Cangül, Aslı Kutlu, Nesrin Özören, Osman Uğur Sezerman.

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
