## [Decision Letter · Decision Letter 0]

26 Aug 2021

PONE-D-21-17970

Computational validation of missense filamin-A mutations, including a novel p.Arg484Gln mutation of two brothers with periventricular nodular heterotopia

PLOS ONE

Dear Dr. Gerlevik,

Thank you for submitting your manuscript to PLOS ONE. After careful consideration, we feel that it has merit but does not fully meet PLOS ONE’s publication criteria as it currently stands. Therefore, we invite you to submit a revised version of the manuscript that addresses the points raised during the review process.

Both reviewers have indicated that the study has merit for publication in PLoS One provided that the minor revisions indicated by both reviewers are addressed.cThere are numerous grammatical errors that need correcting.

We look forward to receiving your revised manuscript.

Kind regards,

Andrew John Sutherland-Smith, Ph.D.

Academic Editor

PLOS ONE

1. Please ensure that your manuscript meets PLOS ONE's style requirements, including those for file naming. The PLOS ONE style templates can be found at https://journals.plos.org/plosone/s/file?id=wjVg/PLOSOne_formatting_sample_main_body.pdf and https://journals.plos.org/plosone/s/file?id=ba62/PLOSOne_formatting_sample_title_authors_affiliations.pdf.

Additional Editor Comments (if provided):

Specific comments:

The following need correcting

Title "Computational validation of missense filamin-A mutations..", the term 'validation' is probably not the best description of the approach taken in the manuscript.

Abstract "However, the effects of these mutations seemed temporary." The use of temporary here is ambiguous, please explain in more detail, or change the term.

Line 253 'Only proteins were kept from the crystals.' Please explain more fully.

For figure 3 it is hard to see where the labelled amino acids exactly are on the structure. Please label the C-alpha atom position to show exactly where the named amino acid is or superimpose a circle on the figure. The labels need to be moved so they are not overlapping the structure for clarity.

Reviewers' comments:

Reviewer's Responses to Questions

**Comments to the Author**

1. Is the manuscript technically sound, and do the data support the conclusions?

Reviewer #1: Yes

Reviewer #2: Yes

2. Has the statistical analysis been performed appropriately and rigorously? 

Reviewer #1: Yes

Reviewer #2: Yes

3. Have the authors made all data underlying the findings in their manuscript fully available?

Reviewer #1: Yes

Reviewer #2: Yes

4. Is the manuscript presented in an intelligible fashion and written in standard English?

Reviewer #1: Yes

Reviewer #2: Yes

5. Review Comments to the Author

Reviewer #1: 1. Authors have identified the variant for FLNa from literature source. Did authors tried to identify the reported mutations from HGMD, dbSNP, and ClinVar databases?

2. What is the reason to choose 4 different PDB structures for this study?

3. Please use and verify the mutation’s pathogenicity and stability by using different tools and compare them with the patient’s phenotypes.

4. The reason behind selecting only p.Arg484Gln mutation is not present in the study. Please mention it.

5. I request authors to compare the significant relationship between the FLNA and FLNB proteins in the discussion section. Please refer the following articles: https://doi.org/10.1002/jcb.25920, https://doi.org/10.3390/molecules25235543.

6. Please remove the Microsoft Word Markup option before submitting the revised article.

7. Discuss the PCA results from MD simulations in detail. What authors have inferred from this analysis?

8. Individual representation of the mutations and native will help to understand the structural insights upon mutation.

9. Spacing errors should be rectified in the manuscript.

Reviewer #2: Gerlevik et al. identify a new R484Q mutation in Filamin A associated with periventricular nodular heterotopia, and the authors investigate the biophysical basis for this disease-causing mutation and others. My background is in protein structure and dynamics, and so I have reviewed the paper in this capacity.

With molecular dynamics simulations, the authors interrogated how the mutations impact the structure and dynamics of FLNa. These insights lead to the identification that the mutations do not globally unfold the proteins but rather subtle affect the motions and tertiary structure around the sites of mutation. I recommend the manuscript for publication after minor revisions, and I have included my comments below.

Page 11 – Line 126: disease mechanism of PNH is reported to be due to loss-of-function of FLNa.Could the authors please comment on the LoF of FLNa caused by these mutations. Do the mutations destabilize FLNa and cause unfolding / aggregation / clearance by the UPS? Or do the mutations simply disrupt protein-protein interactions of FLNa? It would be valuable to know how the predicted in silico stability of FLNa (or the local domain that harbors the mutation of interest) is changed by the mutation. For example, FoldX can be used for this or the webserver MAESTRO.

Page 15 – Line 241: “We also filtered out the variants of which the corresponding amino acids are not determined in any structure submitted to RCSB PDB (77).” Could you please clarify this sentence. Do you mean that you only selected FLNa structures that had the mutation present in the structure? I don’t think this is the case because M102V, S149F, A39E, etc. all have the same PDB ID (3hop). So, does it mean that you only selected FLNa structures that had the wild-type residue of interest in the structure? (in this case, M102, S149, A39, etc.)?

Page 15 – Line 242 Please order the mutations in Table 1 in ascending order (e.g., A39E, L80V, M102V, etc.)

Page 19 – Line 335-336. Could you please elaborate on how the changes were “directly or allosterically related to the PPIs or regulatory functions”? Include specific examples here.

Page 19 - Line 336. What do you mean by “they were mainly temporary”. Does this mean that the mutant structure transiently sampled an alternative conformation, and then reverted back to a “wild-type-like” structure during the simulation?

Page 22 – Line 420-422 How do you know that the mutations do not cause more “general or permanent impairments” [to the FLNa protein]? I assume that you mean the mutants do not globally or partially unfold because your MD simulations showed that the RMSD, RMSF, SASA, etc. were similar enough to the wild-type protein. However, the mutations could affect the stability of the protein and cause it to partially unfold or to be more prone to degradation in vivo by the ubiquitin-proteasome system (UPS). Could you please state why you don’t think the latter options are valid?

Minor comments:

Page 11, line 124 – Pro2641 is listed as a phosphorylation site. Is this correct?

6. PLOS authors have the option to publish the peer review history of their article (what does this mean?). If published, this will include your full peer review and any attached files.

Reviewer #1: No

Reviewer #2: No

---

## [Author Response · Author response to Decision Letter 0]

24 Feb 2022

Point 1. Please ensure that your manuscript meets PLOS ONE's style requirements, including those for file naming. The PLOS ONE style templates can be found at https://journals.plos.org/plosone/s/file?id=wjVg/PLOSOne_formatting_sample_main_body.pdf and https://journals.plos.org/plosone/s/file?id=ba62/PLOSOne_formatting_sample_title_authors_affiliations.pdf. 

Answer 1: We changed the file naming as guided in the PLOS ONE style templates above.

Point 2. Your ethics statement should only appear in the Methods section of your manuscript. If your ethics statement is written in any section besides the Methods, please delete it from any other section.

Answer 2: The ethics statement appears now only at the Methods section of the manuscript.

Point 3. Please review your reference list to ensure that it is complete and correct. If you have cited papers that have been retracted, please include the rationale for doing so in the manuscript text, or remove these references and replace them with relevant current references. Any changes to the reference list should be mentioned in the rebuttal letter that accompanies your revised manuscript. If you need to cite a retracted article, indicate the article’s retracted status in the References list and also include a citation and full reference for the retraction notice.

Answer 3: We reviewed each reference in our list. None of them is retracted, and all of them are correct and complete as far as we analyzed.

Additional Editor Comments (if provided):

Specific comments:

The following need correcting

Comment 1: Title "Computational validation of missense filamin-A mutations..", the term 'validation' is probably not the best description of the approach taken in the manuscript.

Response 1: Thank you. We changed the title as “Computational analysis of missense filamin-A variants, including the novel p.Arg484Gln variant of two brothers with periventricular nodular heterotopia”.

Comment 2: Abstract "However, the effects of these mutations seemed temporary." The use of temporary here is ambiguous, please explain in more detail, or change the term.

Response 2: We are thankful to the editor for highlighting this important issue. As suggested by Reviewer 2, we altered this use of “temporary” with a clearer statement that explains the non-persistence of the observed pathogenic effects during the simulations and probable allowance for the survival of those patients by rescuing the protein’s cell migration initiation.

Comment 3: Line 253 'Only proteins were kept from the crystals.' Please explain more fully.

Response 3: Thank you for raising this point since being clear is very important for us. Thus, we changed this explanation to “Only the atoms of FLNa were kept, the water and smaller molecules related to the crystallization procedure were removed from the crystal structures.” to fully explain this procedure.

Comment 4: For figure 3 it is hard to see where the labelled amino acids exactly are on the structure. Please label the C-alpha atom position to show exactly where the named amino acid is or superimpose a circle on the figure. The labels need to be moved so they are not overlapping the structure for clarity.

Response 4: To improve the clarity of the figure, we relocated the labels outside the Figure 3 and showed the place of the amino acids with arrows as recommended by the editor. We thank to the editor and present the figure for re-evaluation.

Reviewers' comments:

Reviewer's Responses to Questions

Comments to the Author

1. Is the manuscript technically sound, and do the data support the conclusions?

Reviewer #1: Yes

Reviewer #2: Yes

2. Has the statistical analysis been performed appropriately and rigorously?

Reviewer #1: Yes

Reviewer #2: Yes

3. Have the authors made all data underlying the findings in their manuscript fully available?

Reviewer #1: Yes

Reviewer #2: Yes

4. Is the manuscript presented in an intelligible fashion and written in standard English?

Reviewer #1: Yes

Reviewer #2: Yes

5. Review Comments to the Author

Reviewer #1: 

Comment 1: Authors have identified the variant for FLNa from literature source. Did authors tried to identify the reported mutations from HGMD, dbSNP, and ClinVar databases?

Response 1: The variants were annotated and compared to those documented in publicly accessible genetic variant databases (e.g., dbSNP138, ClinVar, ExAC, and gnomAD) using AnnoVar. The available published data in the literature were used to compare the phenotypes of the affected males with the males in this manuscript. We added this information into the manuscript. Thank you very much for the comment.

Comment 2: What is the reason to choose 4 different PDB structures for this study?

Response 2: FLNA is a large protein (>2500 amino acids), and its structure has not been fully resolved by crystallography or any other structural method. That's why we used these four PDB structures that cover the variants we listed. To make it clearer in the manuscript, we mentioned this as one of our limitations in the last paragraph of discussion.

Comment 3: Please use and verify the mutation’s pathogenicity and stability by using different tools and compare them with the patient’s phenotypes.

Response 3: We agree with the reviewer’s comment. Since each tool has several varying strengths and weaknesses, we preferred to consider several pathogenicity predictors and meta-predictors. With the changes we made in the methods part, we tried to make the variant prioritization process clearer. 

Comment 4: The reason behind selecting only p.Arg484Gln mutation is not present in the study. Please mention it.

Response 4: Thank you for the recommendation. A new paragraph is added to the end of the “Identification of a novel variant in FLNA” section of the results, which explains the reason selecting the p.Arg484Gln variant to analyze in detail. Since it is the novel variant we found in the family, we did its analysis in detail, and we used the other variants as validation of our observations.

Comment 5: I request authors to compare the significant relationship between the FLNA and FLNB proteins in the discussion section. Please refer the following articles: https://doi.org/10.1002/jcb.25920, https://doi.org/10.3390/molecules25235543.

Response 5: We added a new paragraph before the last paragraph of our discussion section for the relationship between the FLNA and FLNB by referring the two articles mentioned in the comment.

Comment 6: Please remove the Microsoft Word Markup option before submitting the revised article.

Response 6: We used the markup option to correct the issues requested by the editorial office before the revision procedure. Probably, this is the reason why the first version had markups. We think this is the best way to show our changes upon the reviewers’ and editor’s comments in the revised article. Thus, we used the markup option in the revised article again. However, PLOS ONE also request a clear version of the manuscript in the revision. Therefore, we uploaded both versions to the system for your review. Thanks.

Comment 7: Discuss the PCA results from MD simulations in detail. What authors have inferred from this analysis?

Response 7: Many thanks for raising this point since we missed to give details regarding the PCA results. We now mentioned the conformational changes based on the energetical local minima differences observed in PCA results. We altered and added the related sentences in “Impact of the novel p.Arg484Gln variant” and “Impact of the other variants seen in males as a validation” sections of the results.

Comment 8: Individual representation of the mutations and native will help to understand the structural insights upon mutation.

Response 8: Thanks for this valuable suggestion. However, there are high number of variants that makes a lot of individual figures which would be hard to follow. On the other hand, none of them had a big impact on the structure as mentioned through the manuscript. Therefore, we considered that the representations in Figs 3 and 5 are enough to show that the structure was not disrupted much.

Comment 9: Spacing errors should be rectified in the manuscript.

Response 9: Thank you for noticing these errors. We carefully went over the whole manuscript to fix spacing errors, typos, and redaction flaws.

Reviewer #2:

Gerlevik et al. identify a new R484Q mutation in Filamin A associated with periventricular nodular heterotopia, and the authors investigate the biophysical basis for this disease-causing mutation and others. My background is in protein structure and dynamics, and so I have reviewed the paper in this capacity.

With molecular dynamics simulations, the authors interrogated how the mutations impact the structure and dynamics of FLNa. These insights lead to the identification that the mutations do not globally unfold the proteins but rather subtle affect the motions and tertiary structure around the sites of mutation. I recommend the manuscript for publication after minor revisions, and I have included my comments below.

Comment 1: Page 11 – Line 126: disease mechanism of PNH is reported to be due to loss-of-function of FLNa.Could the authors please comment on the LoF of FLNa caused by these mutations. Do the mutations destabilize FLNa and cause unfolding / aggregation / clearance by the UPS? Or do the mutations simply disrupt protein-protein interactions of FLNa? It would be valuable to know how the predicted in silico stability of FLNa (or the local domain that harbors the mutation of interest) is changed by the mutation. For example, FoldX can be used for this or the webserver MAESTRO.

Response 1: Thank you for this suggestion. We have already performed such an analysis and showed the related results in S1 and S4 Figs (Additional file 1 and 7 with their previous names) as “Binding energy, or complex stability” and “Structural stability” box plots, respectively. However, we missed to mention well to these results within the manuscript. Thus, we thank to the reviewer for raising this point.

The following sentence found in the introduction section, “Disease mechanism of FLNA-associated PNH is known as loss-of-function [3,7,9,21].”, gives an idea about the general mechanism of PNH as it is known in the literature. In this study, one of the main goals is trying to discover how this loss of function happens as you suggested to analyze. Thus, we did molecular dynamics analysis for twelve mutations associated with PNH in males. We also performed structural stability and complex stability analyses by using FoldX on the snapshots from different time points of our simulations. In “S1 Fig” and “S4 Fig”, these results are available. In this distribution-based comparison, there were no significant differences in the mutant structures compared to the wild-type as shown. We altered our results sections to mention these supplementary results (see the sentences referring to S4 Fig and S1 Fig).

Comment 2: Page 15 – Line 241: “We also filtered out the variants of which the corresponding amino acids are not determined in any structure submitted to RCSB PDB (77).” Could you please clarify this sentence. Do you mean that you only selected FLNa structures that had the mutation present in the structure? I don’t think this is the case because M102V, S149F, A39E, etc. all have the same PDB ID (3hop). So, does it mean that you only selected FLNa structures that had the wild-type residue of interest in the structure? (in this case, M102, S149, A39, etc.)?

Response 2: Many thanks for the comment, being clear is important for us. As your last inference, we have only selected the structures that had the wild-type residue of interest in the structure. FLNa is a large protein (>2500 amino acids), and its structure has not been fully resolved by crystallography or any other structural method. Therefore, we used these four PDB structures that cover the variant sites. To make it clearer in the manuscript, we mentioned this as one of our limitations in the last paragraph of discussion.

Comment 3: Page 15 – Line 242 Please order the mutations in Table 1 in ascending order (e.g., A39E, L80V, M102V, etc.)

Response 3: We thank the reviewer for the comment. Table 1 and S1 Table were changed accordingly.

Comment 4: Page 19 – Line 335-336. Could you please elaborate on how the changes were “directly or allosterically related to the PPIs or regulatory functions”? Include specific examples here.

Response 4: We agree with the reviewer, many thanks for this suggestion that made our article more understandable. Thus, we added specific examples for each case at the end of the second paragraph of the “Impact of the other variants seen in males as a validation” section.

Comment 5: Page 19 - Line 336. What do you mean by “they were mainly temporary”. Does this mean that the mutant structure transiently sampled an alternative conformation, and then reverted back to a “wild-type-like” structure during the simulation?

Response 5: Yes, this was what we tried to mean. Thus, we are very thankful to the reviewer for the recommendation of this clear explanation. We altered all similar issues with the usage of “temporary” word with this statement, which makes the manuscript clearer and more understandable.

Comment 6: Page 22 – Line 420-422 How do you know that the mutations do not cause more “general or permanent impairments” [to the FLNa protein]? I assume that you mean the mutants do not globally or partially unfold because your MD simulations showed that the RMSD, RMSF, SASA, etc. were similar enough to the wild-type protein. However, the mutations could affect the stability of the protein and cause it to partially unfold or to be more prone to degradation in vivo by the ubiquitin-proteasome system (UPS). Could you please state why you don’t think the latter options are valid?

Response 6: We agree with the reviewer’s comment about our statement, which seems unclear from this point of view. On the other hand, reviewer’s assumption is correct, and the mutations could affect such global mechanisms. In the MD simulations, we also observed that the FoldX stability of the mutant proteins was close enough to the wild-type’s (S4 Fig), so we can say that the structural stabilities of the domains involved in the PDBs were not influenced by the variants. However, since we performed the MD simulation by using only a certain part of the FLNa protein, we cannot know about more general impairments in the full structure or the dimer complex (even though we partially know for the IgFLNa21 and IgFLNa24 domains, which are significant domains in the dimerization of FLNa). Thus, we modified our statement as below:

“We concluded that the variants seen in the liveborn males result in transient pathogenic effects on the analyzed FLNa domains, rather than persistent impairments that might prevent a partial function, which is required for their survival, with the only copy of FLNA gene.”

Minor comments:

Comment 7: Page 11, line 124 – Pro2641 is listed as a phosphorylation site. Is this correct?

Response 7: Thank you very much for noticing and bringing up this issue. The three residues are not separate phosphorylation sites, and instead, the Ser2640-Pro2641-Tyr2642 region is a putative proline-directed phosphorylation site (with motif Ser/Thr-Pro-X) according to Seo et al. (2009). There is a misunderstanding because of structure of our sentence, which we corrected it as below: 

“However, the putative phosphorylation site, Ser2640-Pro2641-Tyr2642, can be a regulatory switch for these interactions [17].”

Seo, M.-D., Seok, S.-H., Im, H., Kwon, A.-R., Lee, S.J., Kim, H.-R., Cho, Y., Park, D. and Lee, B.-J. (2009), Crystal structure of the dimerization domain of human filamin A. Proteins, 75: 258-263. https://doi.org/10.1002/prot.22336. This site is found and specified as “The putative phosphorylation site, Ser2640-Pro2641-Tyr2642 is also found in carboxy terminus of FLNa24.”

6. PLOS authors have the option to publish the peer review history of their article (what does this mean?). If published, this will include your full peer review and any attached files.

Do you want your identity to be public for this peer review? For information about this choice, including consent withdrawal, please see our Privacy Policy.

Reviewer #1: No

Reviewer #2: No

---

## [Editor Report · Decision Letter 1]

2 Mar 2022

Computational analysis of missense filamin-A variants, including the novel p.Arg484Gln variant of two brothers with periventricular nodular heterotopia

PONE-D-21-17970R1

Dear Dr. Gerlevik,

We’re pleased to inform you that your manuscript has been judged scientifically suitable for publication and will be formally accepted for publication once it meets all outstanding technical requirements.

Kind regards,

Andrew John Sutherland-Smith, Ph.D.

Academic Editor

PLOS ONE

Additional Editor Comments (optional):

A typographical error in Supporting Information Fig 2 needs to be corrected; the label 'Lue656Phe' should be 'Leu656Phe'
---

## [Editor Report · Acceptance letter]

7 Mar 2022

PONE-D-21-17970R1 

Computational analysis of missense filamin-A variants, including the novel p.Arg484Gln variant of two brothers with periventricular nodular heterotopia 

Dear Dr. Gerlevik:

I'm pleased to inform you that your manuscript has been deemed suitable for publication in PLOS ONE. Congratulations! Your manuscript is now with our production department. 

Kind regards, 

on behalf of

Dr. Andrew John Sutherland-Smith 

Academic Editor

PLOS ONE